# Intermittent preventive treatment comparing two versus three doses of sulphadoxine pyrimethamine (IPTp-SP) in the prevention of anaemia in pregnancy in Ghana: A cross-sectional study

**Yaa Nyarko Agyeman** [1]☺*, **Sam Newton** [2‡], **Raymond Boadu Annor** [3‡], **Ellis Owusu-Dabo** [2‡]

**1** Department of Population and Reproductive Health, School of Public Health, University for Development Studies, Tamale, Ghana, **2** Department of Global and International Health, School of Public Health, Kwame Nkrumah University of Science and Technology, Kumasi, Ghana, **3** Savelugu Municipal Hospital, Savelugu, Ghana

☺ These authors contributed equally to this work.
‡ These authors also contributed equally to this work.
* ynyarko@uds.edu.gh

**Data Availability Statement:** All relevant data are within the manuscript and its Supporting Information files.

## Abstract

In 2012 the World Health Organisation (WHO) revised the policy on Intermittent Preventive Treatment with Sulphadoxine Pyrimethamine (IPTp-SP) to at least three doses for improved protection against malaria parasitaemia and its associated effects such as anaemia during pregnancy. We assessed the different SP dosage regimen available under the new policy to determine the dose at which women obtained optimal protection against anaemia during pregnancy. A cross-sectional study was conducted among pregnant women who attended antenatal clinic at four different health facilities in Ghana. The register at the facilities served as a sampling frame and simple random sampling was used to select all the study respondents; they were enrolled consecutively as they kept reporting to the facility to receive antenatal care to obtain the required sample size. The haemoglobin level was checked using the Cyanmethemoglobin method. Multivariable logistic regression was performed to generate odds ratios, confidence intervals and p-values. The overall prevalence of anaemia among the pregnant women was 62.6%. Pregnant women who had taken 3 or more doses of IPTp-SP had anaemia prevalence of 54.1% compared to 66.6% of those who had taken one or two doses IPTp-SP. In the multivariable logistic model, primary (aOR 0.61; p = 0.03) and tertiary education (aOR 0.40; p = <0.001) decreased the odds of anaemia in pregnancy. Further, pregnant women who were anaemic at the time of enrollment (aOR 3.32; p = <0.001) to the Antenatal Care clinic and had malaria infection at late gestation (aOR 2.36; p = <0.001) had higher odds of anaemia in pregnancy. Anaemia in pregnancy remains high in the Northern region of Ghana. More than half of the pregnant women were anaemic despite the use of IPTp-SP. Maternal formal education reduced the burden of anaemia in pregnancy. The high prevalence of anaemia in pregnancy amid IPTp-SP use in Northern Ghana needs urgent attention to avert negative maternal and neonatal health outcomes.

**Funding:** The authors received no specific funding for this work.

**Competing interests:** The authors have declared that no competing interests exist.

## Background

The fundamental aim of malaria interventions in pregnancy is to prevent the adverse effect of malaria in women and the unborn baby [1]. Malaria has been identified as important contributor to high prevalence of anaemia in pregnant women [2, 3]. Malaria-related anaemia compromises maternal health, foetal development and finally result in infant death [4–6]. According to the WHO, haemoglobin concentration below 11.0g/dl defines anaemia during pregnancy [7]. Haemoglobin transports oxygen to body tissues and reductions below acceptable levels has potentially devastating effects on both mother and foetus.

Typically, pregnant women in Ghana are anaemic regardless of malaria infection [6], although anaemia is an important consequence of malaria infection during pregnancy [8, 9]. Although the direct cause of anaemia in African pregnant women are multi-factorial, in malaria endemic regions, placental malaria is responsible for greater proportions of maternal anaemia during pregnancy [9]. Most women are moderately anaemic unlike endemic areas where maternal anaemia is severe, and the supply of routine iron and folic acid supplements have little impact. Meanwhile, malaria-related maternal anaemia affects thiamine levels and results in late infant deaths [10].

The WHO Evidence Review Group recommended the use of at least three (3) SP doses of SP under Directly Observed Therapy (DOT), from the second trimester (first ANC visit) until delivery [11]. The science of the IPTp-SP policy is to protect against maternal parasitaemia [12] and to reduce the risk of malaria-related maternal anaemia [13] so that the negative birth outcomes like low birth weight [12, 14] and stillbirth are prevented [15]. However, there had been several reports of high prevalence of anaemia even after the implementation of the WHO revised IPTp-SP policy [1, 16–20]. Studies show that, 56.0% of pregnant women in low and middle income countries have anaemia [21]. The prevalence of anaemia was high in Nigeria (59.6%) [16], Tanzania; 40.6% [18]; 60.4% [1], Eastern Sudan (62.6%) [17] even after the pregnant women had taken SP. The degree of hemolysis of Red Blood Cells (RBCs) depends on the density of parasites [22].

In Tanzania, where the local transmission of malaria intensity varies regionally, IPTp with SP did not positively impact on maternal anaemia. The prevalence of anaemia in SP-users was 61.0% versus 56.0% among non-SP users in high transmission areas [1]. Hence, pregnant women who did not use IPTp-SP were not disadvantaged in any way in terms of protecting their haemoglobin concentration during pregnancy. McClure *et al*., have reported some inconsistencies of IPTp-SP effectiveness on improving maternal anaemia in a systematic review. Studies evaluating scaled-up programmes found less consistent reductions in anaemia [23]. Likewise, Orish *et.al*., showed in Ghanaian pregnant women that; IPTp-SP intake was not protective against anaemia during pregnancy. The prevalence of anaemia among pregnant women who took IPTp-SP and those who did not take SP was 36.2% and 31.8% respectively. Increasing the SP dose to three or more possesses no potential efficacy to provide meaningful protection against maternal anaemia in pregnancy [24]. All the same, the authors agreed that, more robust study designs could help address the debate about the significance of taking at least 3 doses of SP to control anaemia in pregnancy.

Further, a multi-center study (The Gambia, Benin, and Burkina Faso) found increased SP doses (≥3 doses) remarkably reduced the likelihood of maternal anaemia at delivery though not statistically significant. The published results showed that anaemia prevalence among women who took lower SP doses (less than three) was higher (48–54%) compared to their counterparts (24–29%) who took higher doses (greater than two) [25]. In a metanalysis study, the authors reported that, pregnant women who had taken at least 3 doses had a lower risk of maternal anaemia during pregnancy [26]. In a study done by Kweku and colleagues, they

reported that, the number of SP doses taken during pregnancy provided 57.0% less likelihood of developing any malaria-related anaemia in pregnancy. This dose-dependent effect was however, much prominent in older women compared to their younger counterparts [2]. On the contrary, in Tanzania, anaemia prevalence was high (59.3%) among pregnant women and the odds was three times higher among women with placental malaria infections. Yet, the intake of optimal SP doses during pregnancy showed no significant reduction of maternal anaemia [19].

Ghana initiated the new WHO IPTp-SP recommendation in 2014, and pregnant women are expected to take a minimum of three doses of SP and a maximum of five doses, with each dose taken under the direct observation of a health worker. The first dose is taken at the first ANC enrolment in the second trimester and the remaining doses scheduled to be taken during regular monthly antenatal visits until delivery [27]. The revised policy has been scaled up in all the regions of Ghana including the 26 districts of the Northern Region but its effectiveness in reducing maternal anaemia after implementation is unknown.

Ghana lacks adequate data to either support or rebut the observations, various opinions had been ascribed to debate the issues. The notable ones are whether the effectiveness of the single SP dose was by chance or because many early pregnancy malarias are mild infections, and that the impact of multiple dosing remained undetectable. Additionally, available documents inconclusively adduce that since in practice, women take SP along with folic acid, the practice magnifies SP resistance [28], so that efficacy remains similar irrespective of the dosing frequency. What is missing, pertains to the differential dosing threshold for such resistance or whether the folic acid compromises the effectiveness of the SP or the SP is substandard and rather facilitating the lack of effective treatment.

The different SP doses available under the new programme were assessed to determine the dose under which women obtained optimal protection against maternal anaemia, especially at 36 weeks of gestation.

## Method and materials

### Study design, settings, and participants

A cross-sectional study was conducted enrolling pregnant women who attended antenatal clinic in four selected health facilities (Tamale Teaching Hospital, Tamale West Hospital, Tamale Central Hospital and Tamale Seventh Day Adventist Hospital) in the Tamale metropolis. The register at the facilities served as a sampling frame and simple random sampling was used to select all the study respondents; they were enrolled consecutively as they kept reporting to the facility to receive antenatal care. Pregnant women of at least 16 weeks of gestation, HIV negative, Sickle cell negative and attended antenatal clinic and delivered at the study facilities were included in the study. Pregnant women who migrated out of the study area before the end of the study period and not a resident of the study catchment area was excluded from the study.

Tamale metropolis of the Northern region of Ghana was selected for the study (refer to Fig 1). This region was selectively chosen due to the high level of economic activity in the city of Tamale especially in the Northern belt zone of Ghana. In addition, this region also handles health related issues regarding pregnant women from other satellite areas from the Northern part of the region. The selection of the four sampling sites; namely Tamale Teaching Hospital, Central, West and Seventh Day Adventists Hospitals, was based on high ANC attendance and most ANC attendees return to the facility to give birth during the time of delivery. Moreover, data on pregnant women throughout the course of their pregnancy were readily available.

**Fig 1. Showing the map of Tamale in the Northern region of Ghana.**

The study population was sampled from pregnant women in the Northern region of Ghana who attended antenatal clinic at any of the following hospitals, Tamale Teaching Hospital, Tamale West Hospital, Tamale Central Hospital and Tamale Seventh-Day Adventist Hospital. All pregnant women who qualified for the study were recruited starting from 16 weeks of gestation and/or had experienced quickening.

## Sample size estimation

The target population (N) of the pregnant women was estimated as 64,908 based on the ANC attendance in 2015 from Tamale Teaching Hospital (16,293), Central (22409) West (19,206), and SDA (7000) hospitals. Using a proportion of 85%, which is the reported use of at least one dose of IPTp-SP in 2016 [29] at 95% confidence level and a precision of 3%, assuming a design effect of 2, the sample size obtained was calculated as 1080. To obtain a more precise estimate or statistically significant results, a precision of 3% was assumed, which will increase the sample size as well as the confidence in the results and decrease any uncertainty about the data. We adjusted for a 10% non-response rate and the required sample size was increased to 1188. The number of pregnant women sampled from each study facility was determined by dividing the respective facility antenatal attendance in 2015 with the total attendance of the year (2015) and multiplying the proportion by the study sample size. A quota of 298, 410, 352 and 128 pregnant women were allocated to Tamale Teaching hospital (TTH), Central, West and Seventh Day Adventist hospitals, respectively. The sample size was 1188 pregnant women; however, 1181 respondents were used in the analysis.

## Data collection procedures, instruments, and analysis

A pre-validated questionnaire was developed using extracts from the Multiple Indicator Cluster Survey (MICS), 2011 on maternal and new born health originally designed by United Nations International Children Emergency Fund (UNICEF) [30]. A structured questionnaire was used to interview the pregnant women. The questionnaire was in two phases; the first phase was used to capture information on the pregnant woman during the pregnancy and the second phase was after delivery. It was designed to collect information on socio-demographic characteristics, reproductive history, Insecticide Treated Net (ITN) use, IPTp-SP use, use of anthelmintic drugs and delivery and indicators of health outcome.

Information on reported IPTp-SP use was crosschecked from the Maternal Health record book as well as the register at the clinic. Data was collected at the health facilities during ANC services. The pregnant women were approached during their antenatal care and briefed about the study's objective and characteristics. Eligible participants who consented to the study were selected and interviewed on the first part of the questionnaire which lasted for about 15 to 20 minutes. Each pregnant woman was given a unique code to help identify them during the time of labour and delivery. As per protocol, the haemoglobin level of all pregnant women is checked at the time of ANC registration and 36 weeks of pregnancy. The haemoglobin level at registration (enrollment) was obtained from two main sources: the maternal health record book and the ANC register. Data extracted from the maternal health record book was cross checked with the register at the ANC in accordance with the study protocol and the participants were subsequently interviewed directly to corroborate the validity of all obtained information. At 36 weeks, the haemoglobin level was checked using the Cyanmethemoglobin method. The data collection spanned for a period of 12 months from September 2016 to August 2017.

All data entry and management were conducted using the statistical software, SPSS version 17.0 for Windows (SPSS Inc., Chicago) and transported to STATA 14 for the analysis. During the data analysis, pregnant women who delivered before 36 weeks of gestation were excluded from the analysis. This was because the haemoglobin level could not be checked at 36 weeks of gestation as per protocol.

Categorical variables were compared using Chi-square tests to measure the statistical significance of calculated proportions. At the analysis stage, the participants were stratified into three groups: the no IPTp-SP, <3 doses of IPTp-SP, and ≥3 doses of IPTp-SP. The independent variable for the study was reported usage of IPTp-SP doses and the primary outcome was maternal haemoglobin level. The results obtained from the three study groups were compared and associations were drawn between the doses of IPTp-SP taken and maternal haemoglobin. Univariate and multivariate logistic regressions were performed to generate odds ratios, confidence intervals and a p-value. Variables for the univariate and the multivariate were selected into the model based on the p-value of 0.5 and clinical significance of some risk factors [31, 32]. $P < 0.05$ was considered statistically significant.

## Ethical consideration

Ethical clearance was obtained from the Committee on Human Research, Publication and Ethics of Kwame Nkrumah University of Science and Technology/Komfo Anokye Teaching Hospital before the commencement of the study (CHRPE/AP/375/16). Written consent was obtained from all the pregnant women either by signing or thumbprinting. No minor was used in the study. Permission was obtained from all the four facilities before the commencement of the study. Confidentiality, anonymity, and voluntary participation of participants were ensured during all the phases of data collection.

### Heamoglobin level estimation

The haemoglobin (Hb) of the study women was estimated through the Cyanmethemoglobin method. The method involved pipetting 20 μl of blood of each participant was collected into 4mls of Drapkins solution in a plain glass tube. The Erythrocytes (red blood cells) in the mixture lysed to produce evenly distributed Hb solution. Potassium ferricyanide in the solution converted Hb to methemoglobin which combined with potassium cyanide to form cyanmethemoglobin. All Hbs present in blood were converted to this form. The mixture was placed in a Spectrophotometer and an absorbance was measured at 540nm. Haemoglobin was estimated by comparing the measured absorbance to a corresponding figure on a standard Absorbance/Hb chart [33].

## Results

### Baseline characteristics and anaemia prevalence in late gestation

Table 1 shows the baseline of study participants and anaemia prevalence. Anaemia among women less than 24 years (69.4%) was high compared to pregnant women between the ages of 30 to 34 years old (59.2%). Anaemia decreases with increasing age except among those who were 35 years and above. Generally, anaemia was relatively higher among all the age groups. There was a higher prevalence of anaemia among rural folks (75.0%) compared to urban folks (59.4%). Further, more than half (68.9%) of women without formal education were anaemic as compared to those who had attained college or tertiary education (46.6%). Regarding the occupation of the women, farmers (80.5%) had a higher prevalence of maternal anaemia as compared to salaried workers (43.2%).

There was an association between the trimester at which woman first attended ANC and anaemia in the Pearson chi-square analysis (p<0.001). Those who attended ANC during the first trimester of pregnancy had 56.6% prevalence of anaemia in pregnancy as compared to those who attended ANC in their third trimester (83.8%). However, there was no statistical association between anaemia and parity (p = 0.27) and gravidity (p = 0.88). The prevalence of anaemia among primigravidae (61.2%), secundigravidae (63.3%) and multigravidae (61.8%) were similar as well as nulliparous (61.2%), primiparous (66.9%) and multiparous (61.5%) women in this study (see Table 1).

### Maternal ITN/IPTp-SP use and anaemia

Prevalence of anaemia was high (77.5%) among pregnant women who neither used ITN nor IPTp-SP as compared to those who used some form of intervention during pregnancy. Anaemia prevalence was 63.5%, 61.5% and 58.4% among ITN, SP and both SP and ITN users respectively (Fig 2)

### Prevalence of anaemia in relation to IPTp doses taken

Table 2 below shows the prevalence of anaemia among pregnant women in relation to reported SP use. The overall prevalence of anaemia was 62.6% (in all groups). The prevalence of anaemia among those who did not take SP and those with reported use of SP (at least one dose) was 72.8% and 60.0% respectively.

The prevalence of anaemia decreased with increased usage of SP. Prevalence of anaemia in the no SP group, <3 SP group and ≥3 SP group were 72.8%, 66.6% and 42.4% respectively. It was seen that the prevalence of anaemia was high despite reported usage of IPTp-SP during pregnancy.

**Table 1. Baseline characteristics and prevalence of anaemia in late gestation.**

| Variable | N | | Mean | SD | |
|---|---|---|---|---|---|
| **Age (years)** | **1181** | | **27.1** | **5.93** | |
| **Socio-demographic Variable** | Hb at 36 weeks n (%) 717(62.6) 429(37.4) | | N (1146) | Pearson chi-square | P-value |
| **Age (years)** | Anaemic | Non- Anaemic | | | |
| <24 | 272(69.4) | 120(30.6) | 392 | 13.1043 | **0.004** |
| 25–29 | 226(60.8) | 146(39.2) | 372 | | |
| 30–34 | 135(56.2) | 105(43.8) | 240 | | |
| 35+ | 84(59.2) | 58(40.8) | 142 | | |
| **Marital status** | | | | | |
| Married | 655(61.7) | 406(38.3) | 1061 | 4.2201 | **0.04** |
| Single | 62(72.9) | 23(27.1) | 85 | | |
| **Religion** | | | | | |
| Christian | 63(50.4) | 62(49.6) | 125 | 8.8658 | **0.003** |
| Muslim | 654(64.1) | 367(35.9) | 1021 | | |
| **Residence/locality** | | | | | |
| Rural | 117(75.0) | 39(25.0) | 156 | 13.9633 | **0.001** |
| Peri-urban | 143(64.7) | 78(35.3) | 221 | | |
| Urban | 457 (59.4) | 312(40.6) | 769 | | |
| **Educational level** | | | | | |
| No school | 385(68.9) | 174(31.1) | 559 | 28.1892 | **<0.001** |
| Primary | 151(61.1) | 96(38.9) | 247 | | |
| Secondary | 105(59.3) | 72(40.7) | 177 | | |
| College/tertiary | 76(46.6) | 87(53.4) | 163 | | |
| **Occupation** | | | | | |
| Farmer | 33(80.5) | 8(19.5) | 41 | 51.6898 | **<0.001** |
| Artisan | 143(64.4) | 79(35.6) | 222 | | |
| Salaried worker | 64(43.2) | 84(56.8) | 148 | | |
| Trading | 312(62.2) | 190(37.8) | 502 | | |
| Unemployed | 130(78.8) | 35(21.2) | 165 | | |
| Other | 35(51.5) | 33(48.5) | 68 | | |
| **Ethnicity** | | | | | |
| Dagomba | 556(63.0) | 327(37.0) | 883 | 0.2651 | 0.60 |
| Other ethnicity | 161(61.2) | 102(38.8) | 263 | | |
| **Gravidity** | Positive | Negative | | | |
| Primigravidae | 218(61.2) | 127(36.8) | 345 | 0.2529 | 0.88 |
| Secundigravidae | 164(63.3) | 95(36.7) | 259 | | |
| Multigravidae | 335(61.8) | 207(38.2) | 542 | | |
| **Trimester at 1st ANC** | | | | | |
| First trimester | 280(56.6) | 215(43.4) | 495 | 18.0398 | **<0.001** |
| Second trimester | 406(66.1) | 208(33.9) | 614 | | |
| Third trimester | 31(83.8) | 6(16.2) | 37 | | |
| **Parity** | | | | | |
| Nulliparous | 235(61.2) | 149(38.8) | 384 | 2.5880 | 0.27 |
| Primiparous | 166(66.9) | 82(33.1) | 248 | | |
| Multiparous | 316(61.5) | 198(38.5) | 514 | | |

Note: Data presented as count (percent). Categorical variables compared using Chi-square test and p<0.05 considered statistically significant.

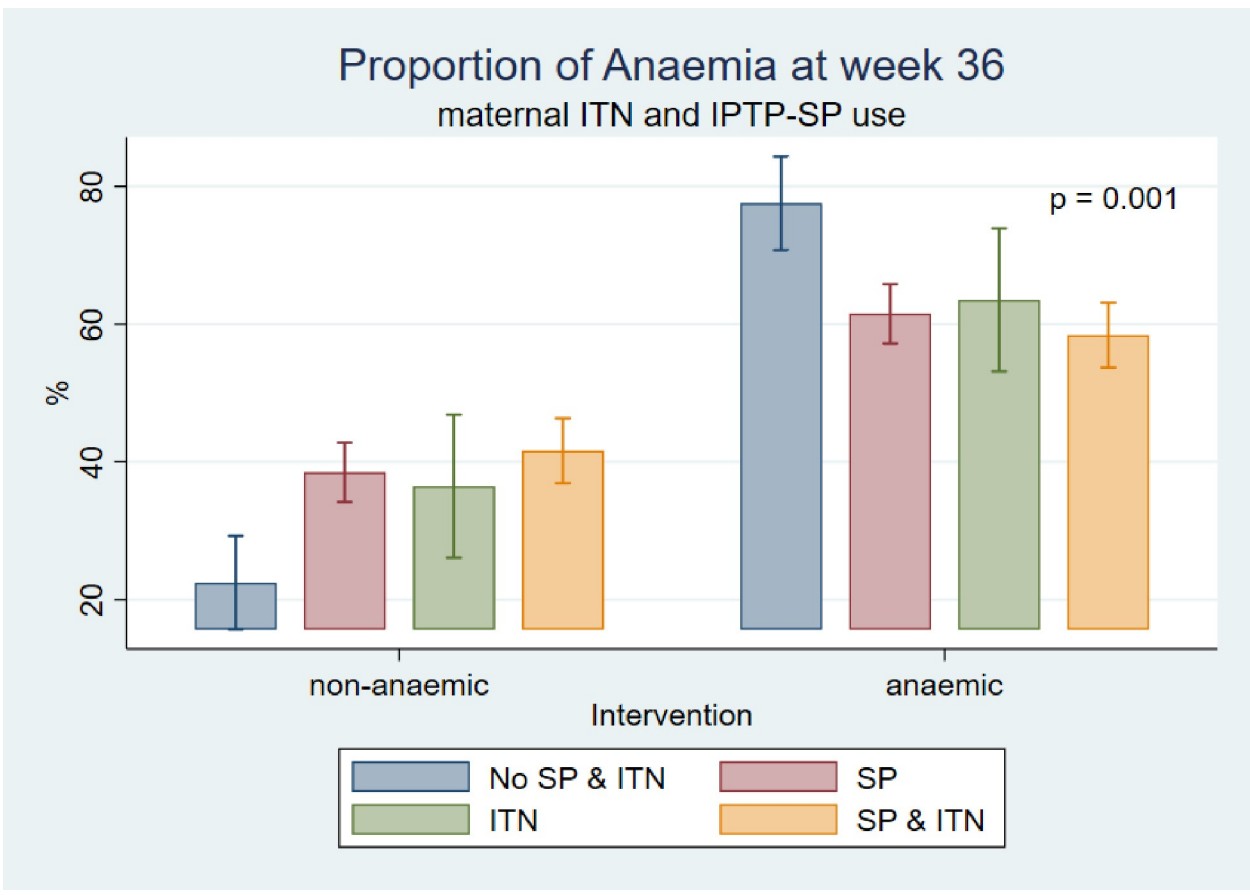

**Fig 2. Maternal ITN/IPTp-SP use and anaemia.**

## Multivariable logistic regression of risk factors and anaemia

Table 3 below shows multivariable logistic regression and risk factors for anaemia in pregnancy. There was no association between age, residence, religion, ethnicity, trimester, and anaemia in pregnancy. Pregnant women who were tertiary graduates were 60% less likely to have maternal anaemia (aOR 0.40; 95%CI 0.25–0.67; p<0.001), compared to those with no formal education. Pregnant women with primary level of education were 39% less likely to have anaemia during their pregnancy (aOR 0.61; 95%CI 0.39–0.95; p = 0.03), compared to those with no education.

**Table 2. Prevalence of anaemia in relation to IPTp doses taken.**

| Variable | anaemia at 36 weeks n (%) | | N (%) | Test statistic | P-value |
|---|---|---|---|---|---|
| | 717(62.6) | 429(37.4) | 1146 | Pearson chi-square | |
| **Reported SP use** | **Yes** | **No** | | | |
| No SP | 169(72.8) | 63(27.2) | 232(100) | **12.0461** | **0.001** |
| Took SP | 549(60.0) | 366(40.0) | 914(100) | | |
| **SP dosage** | | | | | |
| No SP | 169(72.8) | 63(27.2) | 232(100) | **28.2421** | **<0.001** |
| <3 | 285(66.6) | 143(33.4) | 428(100) | | |
| ≥3 | 263(54.1) | 223(45.9) | 486(100) | | |

**Table 3. Multivariable logistic regression of risk factors and anaemia in pregnancy.**

| Variable | Anaemia at 36 weeks | | | | | |
|---|---|---|---|---|---|---|
| | cOR | 95% CI | P-value | aOR | 95% CI | P-value |
| **Age (years)** | | | | | | |
| <25 | 1.00 | Ref | | | | |
| 25–29 | 0.68 | 0.51–0.92 | **0.01** | 0.89 | 0.58–1.37 | 0.6 |
| 30–34 | 0.57 | 0.41–0.79 | **0.001** | 0.69 | 0.43–1.11 | 0.13 |
| ≥35 | 0.64 | 0.43–0.95 | **0.03** | 0.76 | 0.44–1.31 | 0.33 |
| **Residence** | | | | | | |
| Peri-urban | 1.00 | **Ref** | | | | |
| Rural | 1.64 | **1.04–2.58** | 0.034 | 1.17 | 0.62–2.18 | 0.63 |
| Urban | 0.80 | 0.59–1.09 | **0.157** | 0.86 | 0.55–1.32 | 0.48 |
| **Education** | | | | | | |
| No school | 1.00 | **Ref** | | | | |
| Primary | 0.71 | 0.52–0.97 | **0.03** | 0.61 | 0.39–0.95 | **0.03** |
| secondary | 0.66 | 0.46–0.93 | **0.02** | 0.79 | 0.49–1.30 | 0.36 |
| Tertiary | 0.39 | 0.28–0.56 | **<0.001** | 0.40 | 0.25–0.67 | **<0.001** |
| **Religion** | | | | | | |
| Islam | 1.00 | **Ref** | | | | |
| Christianity | 0.58 | 0.39–083 | **0.003** | 1.00 | 0.55–1.86 | 0.98 |
| **Ethnicity** | | | | | | |
| Dagomba | 1.00 | **Ref** | | | | |
| Others | 1.03 | 0.77–1.23 | 0.51 | 0.69 | 0.43.1.11 | 0.13 |
| **Trimester at ANC registration** | | | | | | |
| 1st trimester | 1.00 | **Ref** | | | | |
| 2nd trimester | 1.50 | 1.17–1.91 | **0.001** | 0.97 | 0.68–1.37 | 0.86 |
| 3rd trimester | 3.97 | 1.63–9.68 | **0.002** | 2.85 | 0.77–10.5 | 0.12 |
| **Reported IPTp-SP use** | | | | | | |
| No SP | 1.00 | **Ref** | | 1.00 | | |
| <3 | 0.74 | 0.52–1.06 | **0.098** | 1.03 | 0.63–1.69 | 0.91 |
| ≥3 | 0.44 | 0.31–0.62 | **<0.001** | 0.89 | 0.55–1.45 | 0.65 |
| **ITN use** | | | | | | |
| No | 1.00 | **Ref** | | | | |
| Yes | 0.94 | 0.69–1.27 | 0.68 | 0.93 | 0.66–1.31 | 0.67 |
| **Use of antihelminthics** | | | | | | |
| No | 1.00 | **Ref** | | | | |
| Yes | 0.92 | 0.60–1.42 | 0.71 | 1.48 | 0.82–2.70 | 0.20 |
| **Anaemia at the time of ANC registration** | | | | | | |
| No | 1.00 | **Ref** | | | | |
| Yes | 4.49 | 3.48–5.80 | **<0.001** | 3.32 | 2.36–4.66 | **<0.001** |
| **Malaria infection at the time ANC registration** | | | | | | |
| No | 1.00 | **Ref** | | | | |
| Yes | 0.57 | 0.31–1.06 | 0.08 | 0.78 | 0.34–1.82 | 0.57 |
| **Malaria infection at 36 weeks.** | | | | | | |
| No | 1.00 | **Ref** | | | | |
| Yes | 3.20 | 2.33–4.39 | **<0.001** | 2.36 | 1.56–3.56 | **<0.001** |
| **LBW** | | | | | | |
| ≥2.5kg | 1.00 | **Ref** | | | | |

*(Continued)*

**Table 3.** (Continued)

| Variable | Anaemia at 36 weeks | | | | | |
| --- | --- | --- | --- | --- | --- | --- |
| | cOR | 95% CI | P-value | aOR | 95% CI | P-value |
| <2.5kg | 1.48 | 1.08–2.03 | **0.02** | 1.38 | 0.82–2.32 | 0.22 |

cOR = crude odd ratio, aOR = adjusted odd ratio, 95%CI = 95% confident inferential, p<0.05 considered statistically significant. Ref = 1

The number of doses pregnant women took during pregnancy did not have any effect on maternal anaemia in the adjusted odd ratio. There was no significant association between pregnant women who took three or more doses of SP (aOR 0.89; 95%CI 0.55–1.45; p = 0.65) compared to those who did not take the SP and anaemia in pregnancy. Again, pregnant women who used antihelminthics had no protection against maternal anaemia in pregnancy.

Pregnant women who were anaemic during the time of ANC enrollment were 3.32 times more at risk of anaemia in late gestation (aOR 3.32; 95%CI 2.36–4.66; p<0.001) compared to those who were not anaemic. Similarly, there was an association between pregnant women who had malaria infection in late gestation and anaemia in the multivariable analyses. Pregnant women with malaria infection at 36 weeks of gestation were 2.36 times more at risk of maternal anaemia (aOR 2.36; 95%CI 1.56–3.56; p<0.001) compared to those without malaria infection in late gestation.

## Discussion

### Baseline characteristics and anaemia prevalence in late gestation

The prevalence of anaemia at 36 weeks of gestation was higher in younger maternal age, single women, Muslims, rural residents, illiterates and low- or no-income earners. These findings were consistent with results of a study in Volta region [2], Ashanti region [34] and Northern region [35]. According to Ampofo et al., such socio-demographic characteristics increased the risk of loss to follow-up and subsequent ANC visits, thereby, negatively influencing the adherence level to health interventions applied at the ANC [36]. Potentially, this might have limited the significance of continuous maternity care and increased the risk of anaemia among the women at the later stages of pregnancy. In Northern Ghana, certain occupations (i.e., farming) confine families to reside in the outskirts of communities where access to health care is limited. These families usually have poor nutritional health because their subsistence depends on farm produce which is seasonal compared to the families of salaried workers who have secured supplies of nutritional diversity. Additionally, the initiation and re-attendance of antenatal clinic depended on successful graduation from certain cultural and religious rites performed by the in-laws of pregnant women. For example, the elderly member of the husband's family must publicly declare the pregnancy before women can attend ANC [35]. This might have interrupted the effectiveness of ANC services in reducing the risk of anaemia throughout gestation, especially in illiterates and younger aged women who have had no experience with the use of ITN and IPTp-SP. It was imperative that education on ITN and IPTp-SP be strengthened throughout pregnancy.

The study discovered that, late ANC visit (third trimester) made the women vulnerable to anaemia. This finding is similar to what was found in the study done in Nigeria [37]. However, this finding was contrary to other previous study in Kenya, which found age and occupation as risk factors for anaemia [38]; peripheral and placental malaria, younger age and antenatal care visits were risk factors among Sudanese women [9] and in a study done in Ghana, they found an association between the trimester at first ANC attendance and IPTp-SP use [39]. Early

ANC visit (first trimester) was required to receive the recommended doses (≥3 doses) of IPTp-SP in order to stay protected from anaemia [40].

## Maternal ITN/IPTp-SP and anaemia prevalence

It was observed that, the prevalence of anaemia in pregnancy in late gestation was high among pregnant women, who neither used ITN nor IPTp-SP (77.5%). Anaemia prevalence was high among the respondents who used some interventions; the prevalence was 63.5%, 61.5% and 58.4% among ITN users, SP users and both SP and ITN users, respectively. This was similar with a previous study in Malawi where ITN use did not reduce the risk of maternal anaemia [41]. It was observed that, the prevalence of anaemia was high among pregnant women who only used ITNs or SP and those who combined ITN usage with IPTp-SP. These results might provide empirical evidence in the current study setting that using only the two interventions provided no additive benefit against maternal anaemia in pregnancy. However, in another published study in the Tamale metropolis, the utilisation of ITN or IPTp-SP or both significantly reduced the prevalence of malaria in pregnancy [42]. This suggests that using only IPTp-SP and ITN might still be protective against malaria in pregnancy but not anaemia in Northern Ghana. It is possible that factors such as late gestation malaria infection (as observed in this study), women's knowledge on anaemia [43], food taboos and cultural prohibitions [44], and high-risk nature of the population studied other than helminth infections might be responsible for increased risk of anaemia in pregnancy in this study setting.

## Prevalence of anaemia in relation to IPTp doses taken

The overall prevalence of anaemia in this study (62.6%) was high and in variance with results obtained from previous studies in Benin (14.0–16.3%) [45]. This may be due to the types of study designs, cross-sectioanl versus experimental, which used different sampling methods of recruitment. In our study, respondents were randomly sampled out of the numbers of pregnant women available during each visit whereas Ouedraogo et al., randomized participants based on age and body mass index which are important factors for haemoglobin concentration [45]. Nevertheless, high prevalence of anaemia were recorded in Ho municipality (60.3%) [2], Cape Coast municipality (73.5%) [6], Kassena-Nankana district (47.2%) [46], Sudan (58.9%) [9], and India (72–9%) [47]. These findings suggest that pregnancy-related anaemia is still high in Ghana and the 2014 revised malaria control methods (IPTp with SP) might not provide adequate protection against anaemia in pregnancy in this region. This may confirm the etiology that anaemia in pregnancy is multifactorial that involve other factors like nutrition [1]. Therefore, other S1 Data are required to combat anaemia in pregnancy.

In the current study, the prevalence of anaemia at 36 weeks of gestation was higher (72.8%) among pregnant women who reported no usage of SP compared to those who have taken 1 or 2 doses (66.6%) and three or more doses (54.1%) respectively. These findings are not consistent with results from previous studies in the Sekondi/Takoradi Metropolis of Ghana and Tanzania, where similar anaemia prevalence was observed in both SP users and non-users. In the Ghanaian study, anaemia prevalence was 36.2% among SP users and 31.8% in non-SP users [48]. Likewise, in Tanzania, published ranges of anaemia prevalence in SP users were 46.2–61.0% and 27.3–56.0% in non-SP users. However, our findings were not consistent with earlier works in the Hohoe municipality of Ghana by Kweku et al., [2] and other parts of West Africa [25, 26]. For example, taking additional SP dose more than two significantly reduced malaria parasite density and increased protection against anaemia by 54% [2]. Relatedly, in other parts of West Africa (The Gambia, Benin and Burkina Faso), anaemia prevalence among those who had taken less than three SP doses ranged higher (48–54%) compared to that of three or more

doses (24–29%) [25]. The difference might be due to variations of dosage completion, compromised drug quality, growing drug resistance, variable parasite transmission intensity and weakened pregnancy-specific malarial immunity. In malaria endemic Africa, contrary factors such as poor dosage completion, sub-standard drug quality, increased drug resistance and weakened immunity affect the therapeutic and prophylactic potency of SP. Hence, peripheral blood parasite densities expand and invoke active placental infections late in pregnancy [49, 50]. Other authors have reported in their study, that, IPTp-SP (minimum of three SP doses) use might lower the risk of malaria in pregnancy [1, 42], however, it could not protect pregnant women against anaemia in pregnancy [1]. Though we found a reduction in the prevalence of anaemia in relation to higher reported SP use among the pregnant women, more than half were anaemic despite the use of IPTp-SP. This might hint at growing changes of contrary factors that affect the therapeutic and prophylactic potency of SP administered under IPTp in the country, and also confirm earlier reports of changing malaria epidemiology in Africa [51]. This might have dire implications on anaemia prevention in pregnancy. Additional research is required to profile the impact of the changes on malarial control strategies as well as other pregnancy-related health interventions.

## Multivariable logistic regression of risk factors and anaemia

The current study found education, anaemia at the time of ANC registration and malaria infection in late gestation to be risk factors of anaemia in pregnancy. However, this finding was contrary to other previous studies; in Kenya, age and occupation were risk factors for anaemia [38] and younger age and antenatal care visits were risk factors among Sudanese women [9].

Trimester at which the pregnant women enrolled for ANC had no effect on the prevalence of anaemia in pregnancy in the adjusted odd ratio. However, a study done in Nigeria reported that, late ANC visit made the women vulnerable to anaemia [37]. According to Nkoka et al., early ANC visit (first trimester) was required to receive the recommended doses (≥3 doses) of IPTp-SP in order to stay protected from anaemia [40]. Most of the pregnant women in this study reported to the clinic early when they were in their first and second trimester.

Regardless of the number of doses pregnant women in this setting took, there was no significant association between reported SP usage and maternal anaemia in the adjusted model. The current study findings correlated with a multi-center study done in The Gambia, Burkina Faso, and Benin [25], Sekondi/Takoradi Metropolis of Ghana [48] and in Tanzania [1, 19] where the measured association between higher SP doses (at least three) and dose-dependent protection from maternal anaemia was not statistically significant. This might be indicative of the fact that, using only IPTp-SP could not protect the pregnant women against anaemia. Earlier authors have asserted that improvements of haemoglobin concentration in pregnancy before delivery (particularly around 36 weeks) was not related with IPTp-SP utilization [1, 48].

On the contrary, some published studies demonstrated a dose-dependent potency of IPTp-SP among pregnant women who took variable doses of SP [2, 25, 26]. In a study done by Kweku et al., they reported that, women who had taken at least three SP doses under IPTp were 54% less likely to develop anaemia in pregnancy in the Hohoe municipality [2]. The difference might be related to differences of parasite reduction associated with the consumption of at least three SP doses. For instance, malaria prevalence before IPTp-SP was higher (20.3%) in their study.

So, additional SP doses (beyond two) taken correlated with significant drops in falciparum parasite numbers (18.9%) in their study. Meanwhile, parasite presence did not rapidly destroy red blood cells and drastically reduce haemoglobin concentration to worsen anaemia burden

[52], after the intake of three or more SP doses. In the adjusted model, we did not control for the effect of non-malarial anaemia interventions like iron and folate supplementation in this study. We therefore ignored the influence of non-malarial anaemia interventions in the SP induced pregnant women of the current study participants. However, exposure to antihelminthics, iron and folate supplementation showed negligible impact on maternal anaemia during pregnancy in both SP dosed and non-dosed pregnant women [1]. The current study observed that, maternal formal education, anaemia during early pregnancy and malaria infection during late gestation influence anaemia in pregnancy.

Anaemia in pregnancy in the Northern region remains high despite the reported use of IPTp-SP. Future research may investigate the pathophysiological mechanisms that might have given rise to the high prevalence of anaemia (more than half of the respondents) irrespective of the reported use of IPTp-SP and ITN. Notwithstanding, health planners should implement methods to address inadequate dosage completion, poor drug quality, increasing drug resistance and weakened immunity [19, 50], to reinforce the potency period of higher SP doses (three or more) against peripheral malaria-related anaemia in pregnancy.

## Limitations

The effects of iron and folate supplementations were neither measured nor controlled for at the analysis stage. We relied on documentations in the antenatal register and assumed that participants utilised these interventions in their investigations of IPTp-SP, since they are dispensed during antenatal visitation. Nonetheless, antenatal cards contain factual health care documentations involved with client medical management and moreover, we collected data on ITN use and other known confounders (use of dewormers) and controlled for their effects on the study outcome (anaemia at 36 weeks of gestation).

## Conclusion

Anaemia in pregnancy remains a public health problem in the study area. Anaemia in pregnancy was high among pregnant women in the study setting. Pregnant women with tertiary and primary education had lower odds of anaemia in pregnancy. However, pregnant women who were anaemic at the time of ANC enrollment and had malaria infection at late gestation had higher risk of anaemia in pregnancy. There is an urgent need to address the high prevalence of anaemia in the study setting.

Antenatal care providers should intensify the already existing interventions through education on key areas such as: iron and folate intake from conception of pregnancy until delivery; the need for women to use the already existing malaria prevention intervention strategies and counselling on adequate nutrition and pre-conception care for women in their reproductive age that could prevent malaria infection in late gestation and anaemia at enrollment which were found to increase the odds anaemia in pregnancy. Girl-child education should be encouraged and promoted in the Northern Ghana. Further research is required to undertake a more extensive evaluation of the maternal factors associated with the presence of anaemia among pregnant women.

## Supporting information

**S1 Data.**
(TXT)

**S1 File.**
(DOCX)

## Acknowledgments

We are grateful to the pregnant women who willingly consented to be part of this study. Again, our special thanks go to Dr. Theophilus Adjeso, for proofreading the manuscript and to the hardworking research assistants especially Rita Neindow and Alhassan Bukari. God bless you for the commitment and sacrifice you put in during the data collection. The technical assistance of the midwives at all the data collection centers, the laboratory technicians, and the staff of the health facilities involved in the study are gratefully acknowledged.

## Author Contributions

**Conceptualization:** Yaa Nyarko Agyeman.

**Data curation:** Yaa Nyarko Agyeman, Ellis Owusu-Dabo.

**Formal analysis:** Raymond Boadu Annor.

**Methodology:** Yaa Nyarko Agyeman, Sam Newton.

**Project administration:** Yaa Nyarko Agyeman.

**Resources:** Yaa Nyarko Agyeman.

**Supervision:** Sam Newton, Ellis Owusu-Dabo.

**Writing – original draft:** Yaa Nyarko Agyeman.

**Writing – review & editing:** Sam Newton, Raymond Boadu Annor, Ellis Owusu-Dabo.

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
