## [Decision Letter · Decision Letter 0]

17 Nov 2020

PONE-D-20-28114

Intermittent Preventive Treatment Comparing Two Versus Three Doses of Sulphadoxine Pyrimethamine (IPTp-SP) in the Prevention of Anaemia in Pregnancy in Ghana: A Cohort Study

PLOS ONE

Dear Dr. Agyeman,

Thank you for submitting your manuscript to PLOS ONE. After careful consideration, we feel that it has merit but does not fully meet PLOS ONE’s publication criteria as it currently stands. Therefore, we invite you to submit a revised version of the manuscript that addresses the points raised during the review process.

As is clear from the expert reviewers' comments, with which I entirely agree, there are fundamental issues, in particular with the study design, methodology and statistical analytical approach, that must by obligation be addressed point-by-point in any revised manuscript that you intend to submit for re-consideration. Along with these substantial changes, the numerous typographical and grammatical errors need also to be eliminated.

We look forward to receiving your revised manuscript.

Kind regards,

Adrian J.F. Luty, PhD

Academic Editor

PLOS ONE

Journal Requirements:

2. We note you have included a table to which you do not refer in the text of your manuscript. Please ensure that you refer to Table 5 in your text; if accepted, production will need this reference to link the reader to the Table.

4. In the Methods, please clarify that participants provided oral consent. Please also state in the Methods:

- Why written consent could not be obtained

- Whether the Institutional Review Board (IRB) approved use of oral consent

- How oral consent was documented

For more information, please see our guidelines for human subjects research: https://journals.plos.org/plosone/s/submission-guidelines#loc-human-subjects-research

5. In your Methods section, please provide additional information about the participant recruitment method and the demographic details of your participants. Please ensure you have provided sufficient details to replicate the analyses such as: a) a description of any inclusion criteria that were applied to participant recruitment (the current manuscript lists the exclusion criteria twice), b) a statement as to whether your sample can be considered representative of a larger population, c) a description of how participants were recruited, and d) descriptions of where participants were recruited and where the research took place.

6. You indicated that you had ethical approval for your study. In your Methods section, please ensure you have also stated whether a) if minors were included in your sample, and if so, b) you obtained consent from parents or guardians of the minors included in the study or whether the research ethics committee or IRB specifically waived the need for their consent.

Reviewers' comments:

Reviewer's Responses to Questions

**Comments to the Author**

1. Is the manuscript technically sound, and do the data support the conclusions?

Reviewer #1: Partly

Reviewer #2: No

2. Has the statistical analysis been performed appropriately and rigorously? 

Reviewer #1: No

Reviewer #2: No

3. Have the authors made all data underlying the findings in their manuscript fully available?

Reviewer #1: Yes

Reviewer #2: Yes

4. Is the manuscript presented in an intelligible fashion and written in standard English?

Reviewer #1: No

Reviewer #2: No

5. Review Comments to the Author

Reviewer #1: General comments:

This is a manuscript which objective was to compare two versus three doses of Intermittent Preventive Treatment (IPTp) of Sulfadoxine-pyrimethamine (SP) in preventing anemia in pregnant women in Ghana, by Yaa Nyarko et al.

Although the manuscript objective is relevant, some caveats are found in the methods, the results and the conclusion sections as suggested below:

1. Abstract:

Line 25: It is said that “….from a maximum of two…” Actually this statement is not correct, as WHO recommended at least two doses and not only two doses. Thus, the maximum of two is not accurate.

Line 41: As, this was a cohort study, one is wondering if odds ratio should be used as a measure of the estimate. More details are provided the method section.

2. Introduction:

Line 58: a reference is needed for this sentence. Also, this first sentence should be given after the second sentence which should come first.

Line 66-67. This sentence is confusing as anaemia in pregnant women is multifactorial, but not saying that “ direct causes of anaemia … are multifactorial…” Also, poverty is not a direct cause of anaemia. Thus, this sentence needs to be clearly written.

Line 93-95: the sentence “they further…….. in pregnancy” is confusing as written. Thus, a clear sentence is needed.

3. Methods:

Line 143: How women in the no IPTp-SP group were followed until 36 weeks from their inclusion. An ethical issue could happen in the context of this study.

Line 144: It is unclear how IPTp-SP doses is considered as a depending variable.

Line 151: is that mean that all women were recruited at 16 weeks gestation? This needs be clarified, as inclusion criteria below is not clear.

Line 154-156: Are these elements part of the inclusion criteria. I wonder if these are not exclusion criteria instead. Thus, the elements of the inclusion criteria are expected.

Line 167: A precision of 3% was used in the sample size calculation. Can the authors justify why this was used in parallel with high proportion (85%).

Line 197: Univariate and multivariate analysis were used to measure our estimate. Was it a logistic regression? Also, as this was a follow-up study, one would wonder if an odds ratio should be used. In addition, one would like to know how variable in the univariate and multivariate models have been selected, as this could change the direction of the results generated in the two models as well as the conclusion provided.

4. Results:

Line 205: “Overall……in the study” in fact, all the 1188 were included initially, but not only the 1181 whom completed the follow-up. This in the baseline characteristics suggested in table 1, results should reflect the 1188 and not only the 1181. Also, in table 1, where ever mean is reported, a standard deviation should follow. Eg. Age….

Table 2:

Trimester at 1st ANC: As inclusion criteria was not stated correctly, one would wonder how women included from 25 to term were assessed for their haemoglobin at 36 weeks gestation. Please clarify that.

Line 247: “the was…..current study” there is a typos error here to check.

Table 5: Among the variables included in this model, one would like to see the inclusion of other variables such as anaemia at enrolment, trimester or gestational age of SP uptake, ITN use, ect… Authors should provide the criteria for a variable to be included in the two models. With the absence of important variables in the models, one would be cautious about the conclusion provided for this study. Thus, the discussion section could be adjusted accordingly.

Table2: Less than 3 doses where protective of anaemia in the univariate model although borderline significant. However, in the multivariate model, this is considered as a risk factor. Can the authors provide this justification?

Conclusion:

In this section, results in term of numbers are not expected. Eg 55.8%. 62.6%, ect…. These are already available in the results section. Also, explanation (eg line 386-line 388) is not supposed to be given in that section as discussion section is allocated for that.

Line 391-392: The conclusion could be reviewed after controlling for other additional variables in the regression model.

Reviewer #2: The purpose of this article is to test the effectiveness of the new policy on intermittent preventive treatment with Sulfadoxine-Pyrimethamine in Ghana to prevent anemia in pregnant women, and to determine the optimal dosage of protection for pregnant women against anemia.

The subject is not without interest, but the manuscript in its current form suffers from major drawbacks which seriously compromise the reliability of the conclusions drawn by the authors:

The study is presented as a cohort study but it is not clear how the data were processed longitudinally. The main outcome is anemia at 36 weeks gestation, which is a very partial summary of anemia during pregnancy, and the choice of this outcome is not explained

It is not clear whether pregnant women are recruited at the first antenatal visit or later during pregnancy.

The method of measuring hemoglobin level is not mentioned.

It is not mentioned if some women received doses of IPT after 36 weeks of pregnancy (which would of course compromise the conclusion about the impact of IPTp on anemia at 36 weeks)

No comparison between women who received less than 2 doses and those who received 3 or more doses was shown, which makes it difficult to assess the potential study biases.

The manuscript presents the calculation of a sample size without the assumption on which this calculation is based being clearly explained, and without any evident link to the stated objectives of the article.

Table 4 is presented as a multivariate logistic regression but no details are given neither on the adjustment variables nor on the method of variable selection

The authors mention a design-effect (intra-hospital correlation) without showing how the statistical analysis takes this element into account.

The draft does not seem to have been reread in a rigorous manner, and several inconsistencies remain:

- the paragraphs inclusion criteria and exclusion criteria are identical (and only concern the exclusion criteria),

- line 244 Table 4 is confused with table 5

- etc…

In sumary, the level of evidence of the conclusions is very low.

The authors are encouraged to rework the study methodology to address the significant gaps mentioned above, as well as the data analysis and presentation of results.

6. PLOS authors have the option to publish the peer review history of their article (what does this mean?). If published, this will include your full peer review and any attached files.

Reviewer #1: No

Reviewer #2: No

---

## [Author Response · Author response to Decision Letter 0]

29 Jan 2021

All the comments raised by the reviewers have been addressed in the revised manuscript.

---

## [Decision Letter · Decision Letter 1]

3 Mar 2021

PONE-D-20-28114R1

Intermittent Preventive Treatment Comparing Two Versus Three Doses of Sulphadoxine Pyrimethamine (IPTp-SP) in the Prevention of Anaemia in Pregnancy in Ghana: A Cohort Study

PLOS ONE

Dear Dr. Agyeman,

Thank you for submitting your manuscript to PLOS ONE. After careful consideration, we feel that it has merit but does not fully meet PLOS ONE’s publication criteria as it currently stands. Therefore, we invite you to submit a revised version of the manuscript that addresses the points raised during the review process.

Specifically, both the reviewer and I find that your revised manuscript is improved but there remain issues raised in the first review round that are unanswered. You must address all the issues raised in a further revision.

We look forward to receiving your revised manuscript.

Kind regards,

Adrian J.F. Luty, PhD

Academic Editor

PLOS ONE

Journal Requirements:

Reviewers' comments:

Reviewer's Responses to Questions

**Comments to the Author**

1. If the authors have adequately addressed your comments raised in a previous round of review and you feel that this manuscript is now acceptable for publication, you may indicate that here to bypass the “Comments to the Author” section, enter your conflict of interest statement in the “Confidential to Editor” section, and submit your "Accept" recommendation.

Reviewer #1: (No Response)

2. Is the manuscript technically sound, and do the data support the conclusions?

Reviewer #1: Partly

3. Has the statistical analysis been performed appropriately and rigorously? 

Reviewer #1: No

4. Have the authors made all data underlying the findings in their manuscript fully available?

Reviewer #1: Yes

5. Is the manuscript presented in an intelligible fashion and written in standard English?

Reviewer #1: Yes

6. Review Comments to the Author

Reviewer #1: General comments:

This is a very good and well written manuscript which aims to compare two versus three doses of intermittent preventive treatment using sulfadoxine-pyrimethamine to prevent maternal anaemia during pregnancy. However, there is a need of clarifying some points in order to increase the quality of this work.

1. Abstract:

Line22-23: In fact, this recommendation started from 2012 with a policy brief but not in 2014 as indicated by the authors.

2. Background:

Line 72-74: The sentence “The science of IPTp……prevented” is not from the author indicated in the manuscript. Please make necessary correction

Line 77: The authors indicated that 60% of anaemia was observed in studies from references 1, 14-16 in Tanzania. After verification of the references, the 60% is not found. Please verify or rewrite this sentence according to the reality.

Line 82-83: the sentence “The prevalence…. transmission areas” can the authors indicate the reference of this? Also, is that sentence correct? One would wonder if the authors want to say 61% among non-users versus 56% among users.

3. Methods:

Sample size calculation

Line 161-162: Can the authors provide the rational of using the precision of 3% given the high proportion of 85% considered.

Also, not sure if the proportion of 85% in anticipated target should be used. Instead, proportion is supposed to come from previous studies.

Line 196: as malaria seasonality has been considered, performing the analysis by season is preferable to take this component into account.

Line 207-209: As this was a cohort study according to the authors, one would wonder if Odds ratio should be used to estimate the association between predictors and outcome. Can the authors provide argument about not using risk ratio?

4. Results:

Line 271-272: the sentence: “prevalence of …….pregnancy”. Can the authors provide the over form of intervention used during pregnancy?

Table 3: Can the authors provide the meaning of MPs under the table?

5. Discussion:

Line 353: The sentence: “It was discovered…” Is this a discover? It is probably an observation.

Line 355: The rates of anaemia (63.5% versus 61.5%) provided are said to be different by the authors. One would wonder if this is correct, as these are really similar.

Line 363: The authors suggested that other factors than malaria may be responsible of increasing the risk of anaemia. Can the authors provide these factors?

Line 374-375: As 62% is different from 73.5% and 47.2%, what is the supportive argument of this consistency suggested?

Line 381-383: The prevalence of anaemia provided in different doses of SP seem to indicate a protection against anaemia. However, the conclusion provided by the authors suggest non protective effect of SP on anaemia. Can the authors provide argument to explain this discrepancy between their conclusion and what is observed here (line 381-383)?

Line 412-413: It is said that malaria in this setting is not the cause of anaemia. This statement is questionable, as malaria can always cause anaemia, although it is not main contributor in this setting.

Line 470: limitations

One limitation in this study could be the fact that no parasitemia results were available at 36 weeks to better explain the relationship between IPTp-SP and anaemia.

Conclusion:

Line 480-481: The two sentences can be combined in one.

Line 488-489: “….the need ……the odds of anaemia in pregnancy…”. This is not demonstrated in this study.

Line 491-492: The sentence “further research….part if Ghana” as this is a recommendation, it is important to provide details on the deficiencies related with the intervention used.

7. PLOS authors have the option to publish the peer review history of their article (what does this mean?). If published, this will include your full peer review and any attached files.

Reviewer #1: No

---

## [Author Response · Author response to Decision Letter 1]

30 Mar 2021

University for Development Studies

School of Public Health 

Tamale 

30th March 2021

Dear Editor

Revision was requested to be made regarding a paper that was submitted to your journal with the topic “Intermittent Preventive Treatment Comparing Two Versus Three Doses of Sulphadoxine Pyrimethamine (IPTp-SP) in the Prevention of Anaemia in Pregnancy in Ghana: A Cross-sectional study (Long Title)”

We have addressed the concerns raised by the reviewer. 

We thank the reviewers for the generous comments, their time and expertise and have edited the manuscript to address their concerns as shown below.

Reviewer #1: General comments:

This is a very good and well written manuscript which aims to compare two versus three doses of intermittent preventive treatment using sulfadoxine-pyrimethamine to prevent maternal anaemia during pregnancy. However, there is a need of clarifying some points in order to increase the quality of this work.

1. Abstract:

Line22-23: In fact, this recommendation started from 2012 with a policy brief but not in 2014 as indicated by the authors.

This correction has been done.

2. Background:

Line 72-74: The sentence “The science of IPTp……prevented” is not from the author indicated in the manuscript. Please make necessary correction

The author has changed.

Line 77: The authors indicated that 60% of anaemia was observed in studies from references 1, 14-16 in Tanzania. After verification of the references, the 60% is not found. Please verify or rewrite this sentence according to the reality.

The 60% was for Nigeria. No percentage was stated for Tanzania; however, we have included the prevalence for Tanzania now.

Line 82-83: the sentence “The prevalence…. transmission areas” can the authors indicate the reference of this? Also, is that sentence correct? One would wonder if the authors want to say 61% among non-users versus 56% among users.

These corrections have been made

3. Methods:

Sample size calculation

Line 161-162: Can the authors provide the rational of using the precision of 3% given the high proportion of 85% considered.

This is the reason for using a precision of 3%. This is part of a bigger study and one of the outcomes that was of interest was stillbirth, which is a rare occurrence. The sample size was increased to help get the needed information as well and to help in generalization of the study as well. 

Also, not sure if the proportion of 85% in anticipated target should be used. Instead, proportion is supposed to come from previous studies.

This has been corrected and referenced appropriately.

Line 196: as malaria seasonality has been considered, performing the analysis by season is preferable to take this component into account.

This statement has been corrected.

Line 207-209: As this was a cohort study according to the authors, one would wonder if Odds ratio should be used to estimate the association between predictors and outcome. Can the authors provide argument about not using risk ratio?

Thank you for the comment. The design used has been corrected. 

4. Results:

Line 271-272: the sentence: “prevalence of …….pregnancy”. Can the authors provide the over form of intervention used during pregnancy?

The statement was not too clear however, the other interventions have been included

Table 3: Can the authors provide the meaning of MPs under the table?

It has been written in full, Malaria parasites

5. Discussion:

Line 353: The sentence: “It was discovered…” Is this a discover? It is probably an observation.

It has been changed to an observation

Line 355: The rates of anaemia (63.5% versus 61.5%) provided are said to be different by the authors. One would wonder if this is correct, as these are really similar.

This correction has been done

Line 363: The authors suggested that other factors than malaria may be responsible of increasing the risk of anaemia. Can the authors provide these factors?

The other factors have been provided.

Line 374-375: As 62% is different from 73.5% and 47.2%, what is the supportive argument of this consistency suggested?

The grammar has been corrected

Line 381-383: The prevalence of anaemia provided in different doses of SP seem to indicate a protection against anaemia. However, the conclusion provided by the authors suggest non protective effect of SP on anaemia. Can the authors provide argument to explain this discrepancy between their conclusion and what is observed here (line 381-383)?

This has been done please

Line 412-413: It is said that malaria in this setting is not the cause of anaemia. This statement is questionable, as malaria can always cause anaemia, although it is not main contributor in this setting.

This has been corrected

Line 470: limitations

One limitation in this study could be the fact that no parasitemia results were available at 36 weeks to better explain the relationship between IPTp-SP and anaemia.

Parasite results were available and were part of the adjusted model. That was the MPs…I have written it in full. The effect of the malaria infection was considered, and it was found that, malaria during late gestation significantly influenced anaemia.

Conclusion:

Line 480-481: The two sentences can be combined in one.

It has been done

Line 488-489: “….the need ……the odds of anaemia in pregnancy…”. This is not demonstrated in this study.

it has been corrected.

Line 491-492: The sentence “further research….part if Ghana” as this is a recommendation, it is important to provide details on the deficiencies related with the intervention used.

Thais has been done. 

We believe the manuscript is now ready for publication in the PLOS ONE journal.

Yaa Nyarko Agyeman (Ph.D)

Lecturer 

Dept of Population and Reproductive Health

---

## [Editor Report · Decision Letter 2]

6 Apr 2021

Intermittent Preventive Treatment Comparing Two Versus Three Doses of Sulphadoxine Pyrimethamine (IPTp-SP) in the Prevention of Anaemia in Pregnancy in Ghana: A Cohort Study

PONE-D-20-28114R2

Dear Dr. Agyeman,

We’re pleased to inform you that your manuscript has been judged scientifically suitable for publication and will be formally accepted for publication once it meets all outstanding technical requirements.

Kind regards,

Adrian J.F. Luty, PhD

Academic Editor

PLOS ONE
---

## [Editor Report · Acceptance letter]

8 Apr 2021

PONE-D-20-28114R2 

Intermittent Preventive Treatment Comparing Two Versus Three Doses of Sulphadoxine Pyrimethamine (IPTp-SP) in the Prevention of Anaemia in Pregnancy in Ghana: A Cross-sectional Study  

Dear Dr. Agyeman:

I'm pleased to inform you that your manuscript has been deemed suitable for publication in PLOS ONE. Congratulations! Your manuscript is now with our production department. 

Kind regards, 

on behalf of

Dr. Adrian J.F. Luty 

Academic Editor

PLOS ONE